# Heterogeneous Medical Data Integration with Multi-Source StyleGAN

**Wei-Cheng Lai**[1]                                                    WEI-CHENG.LAI@HPI.DE
**Matthias Kirchler**[1,5]                                          MATTHIAS.KIRCHLER@HPI.DE
**Hadya Yassin**[1]                                                     HADYA.YASSIN@HPI.DE
**Jana Fehr**[1,2]                                                 JANA.FEHR@BIH-CHARITE.DE
**Alexander Rakowski**[1]                                        ALEXANDER.RAKOWSKI@HPI.DE
**Hampus Olsson**[3]                                           N.HAMPUS.OLSSON@GMAIL.COM
**Ludger Starke**[3]                                            LUDGER.STARKE@MDC-BERLIN.DE
**Jason M. Millward**[3]                                       JASON.MILLWARD@MDC-BERLIN.DE
**Sonia Waiczies**[3]                                          SONIA.WAICZIES@MDC-BERLIN.DE
**Christoph Lippert**[1,4]                                         CHRISTOPH.LIPPERT@HPI.DE

[1] *Digital Health and Machine Learning, Hasso-Plattner-Institute, University of Potsdam, Germany*
[2] *QUEST Center for Responsible Research, Berlin Institute of Health, Charité Universitätsmedizin Berlin, Germany*
[3] *Max-Delbrück-Center for Molecular Medicine in the Helmholtz Association (MDC), Berlin Ultra-high Field Facility, Berlin, Germany*
[4] *Hasso Plattner Institute for Digital Health at Mount Sinai, Icahn School of Medicine at Mount Sinai, New York, NY, United States of America*
[5] *University of Kaiserslautern-Landau, Germany*

**Editors:** Accepted for publication at MIDL 2024

## Abstract

Conditional deep generative models have emerged as powerful tools for generating realistic images enabling fine-grained control over latent factors. In the medical domain, data scarcity and the need to integrate information from diverse sources present challenges for existing generative models, often resulting in low-quality image generation and poor controllability. To address these two issues, we propose Multi-Source StyleGAN (MSSG). MSSG learns jointly from multiple heterogeneous data sources with different available covariates and can generate new images controlling all covariates together, thereby overcoming both data scarcity and heterogeneity. We validate our method on semi-synthetic data of handwritten digit images with varying morphological features and in controlled multi-source simulations on retinal fundus images and brain magnetic resonance images. Finally, we apply MSSG in a real-world setting of brain MRI from different sources. Our proposed algorithm offers a promising direction for unbiased data generation from disparate sources. For the reproducibility of our experimental results, we provide detailed code implementation [1].
**Keywords:** Generative Models, StyleGAN, Multi-Source, MRI, Retinal Fundus Images.

## 1. Introduction

Many medical Deep Learning (DL) applications suffer from algorithmic biases, stemming from training with limited and biased data sets due to data restriction and low disease prevalence (Kazeminia et al., 2020; Bak et al., 2022). Currently, research has employed deep

---

1. https://github.com/weslai/msstylegans

generative models to create synthetic Computer Tomography scans, brain Magnetic Resonance Images (MRIs) (Frid-Adar et al., 2018; Han et al., 2018; Hong et al., 2021; Xiang et al., 2023) and retinal fundus images (Costa et al., 2017, 2018; Zhao et al., 2017). This offers an opportunity to fill the data availability gap in the medical domain by augmenting available data with synthetic counterparts (Kazeminia et al., 2020). These applications generate high-resolution medical images but are not able to generate images with specific demographic and clinical characteristics. Other generative approaches use conditional Variational Autoencoders (VAEs) that learn causal relationships to infer brain MRIs with specific clinical and demographic characteristics (Reinhold et al., 2021; Jung et al., 2021) and generate counterfactual brain MRIs (Pawlowski et al., 2020).

In the medical domain, several crucial limitations arise. Generally, data sets are *small*, compared with natural image data sets, different studies present *diverging demographic and distributional stratifications*, and different sources release *different sets of latent factors*. Consider the example of building a generative model to synthesize brain MRIs for developing models that predict dementia. One data source may be a cohort study and collect data from a general population, with age at collection sampled uniformly between 40 and 70; due to the cohort design, Cognitive Impairment (CI) has a very low prevalence in this source. Another data source, on the other hand, may be a specific Alzheimer's Disease (AD) cohort with a very high prevalence of CI in the sampled individuals; since age is a major risk factor for AD, this second data source will likely also collect older individuals. In addition, the two studies' collection protocols will be

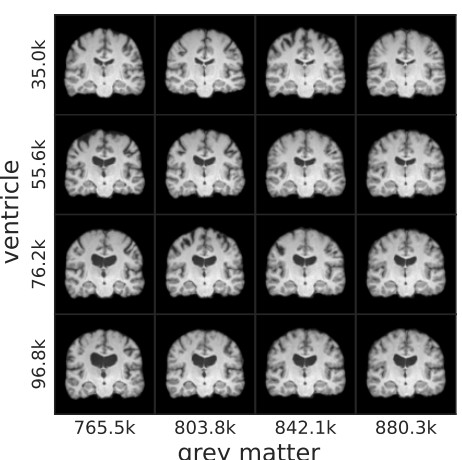

**Figure 1:** Conditionally generated MRI from the multi-source model. Age is fixed at 59. The model controls ventricle volumes and grey matter volumes (white part of the brain MRI) simultaneously.

differently designed to satisfy different aims. Hence, the available covariates for each study will only partially overlap. While both studies are highly likely to collect information on age, demographics, and sex, the first study might collect additional information on other neurological issues, while the second study might collect clinical measurements specific to AD. Indeed, this example is the very case for the UK Biobank (UKB), a UK-based population study (Sudlow et al., 2015), and the Alzheimer's Disease Neuroimaging Initiative (ADNI) (Petersen et al., 2010). Generally, due to privacy concerns and high acquisition costs, medical images are usually also released only in smaller studies and can not be scraped at the same scale as natural images. For example, the UKB contains one of the largest brain MRI studies to date and only acquired images of less than 50,000 individuals so far.

Training reliable generative models on medical imaging data is challenging due to several factors. Limited data availability often leads to lower visual quality and incomplete coverage of the image space. Additionally, the scarcity of labels restricts control over generated images to the covariates present in a single dataset. Most existing conditional generative models cannot leverage multiple data sources simultaneously unless label overlap is perfect

(Han et al., 2021). Moreover, models trained on a single data source suffer from reduced data availability. To address these limitations, we introduce Multi-Source StyleGAN (MSSG), which is based on StyleGAN3 (Karras et al., 2021) but learns from multiple data sources concurrently. This approach increases dataset size and enables conditional image generation with *all* available latent covariates.

We first validate MSSG on a semi-synthetic data set of hand-written digits in which we can directly control specific morphological characteristics (Castro et al., 2019). We also investigate MSSG's behavior on real-world medical imaging data from a single source with a simulated data source split. Finally, we apply MSSG to the realistic setting of multiple sources of brain MRI data. We show that MSSG can synthesize high-quality medical images and jointly control latent structures present in those images.

## 2. Multi-Source StyleGAN

### 2.1. Conditional StyleGANs

Conditional Generative Adversarial Networks (GANs) generate natural images with specified categories (Mirza and Osindero, 2014). Given the training data and labels, the labels are passed to the generator and the discriminator. In the generator, labels are concatenated with noise into a latent representation. In the discriminator, labels are fitted as input and the discriminator distinguishes synthetic images from real images based on labels. For conditional StyleGANs, the generator transforms a one-hot-vector label $c$ into an embedding vector. This is passed to the mapping network $M$ with a latent vector $z$. The mapping network produces another latent vector $w$ and this latent vector $w$ is passed to the synthetic network for image generation. The StyleGAN discriminator applies a conditional projection discriminator (Miyato and Koyama, 2018); see Section A for details.

**Mixed-type latent-variable conditional GANs**   In this work, we focus on an extension of conditional GANs that can (i) handle mixed-type latent factors and (ii) integrate data sources that have different labels available. Here, the generator network takes in both a noise variable $z \in \mathbb{R}^{d_z}$ and a conditioning variable $c \in \mathbb{R}^{d_c}$, which may consist of binary, multiclass (coded as a one-hot subvector), and continuous values. Analogously to the standard StyleGAN definition, the generator maps $c$ through an embedding network and concatenates them to the noise variables $z$ to generate the image. The discriminator predicts $1+d_c$ output variables, with the first output denoting the standard fake/real prediction. The remaining $d_c$ outputs are predictions for each of the input variables, with appropriate loss functions for each variable (e.g., the cross-entropy loss for categorical subfeatures, and the quadratic loss for continuous subfeatures). This requires a generative model of the latent distribution, which will be learned independently from the covariates, as described in the next section.

### 2.2. Modeling of Latent Factors

For simplicity, we will focus on the case of two sources, $D^1$ and $D^2$. Each data source $j$ consists of image-covariate pairs, $D^j = \left\{ (x_i^j, c_i^j) \right\}_i$. The covariates consist of a *shared part* $c_{\mathrm{shar}}^j$ that is available in both sources, a *unique part* $c_{\mathrm{uniq}}^j$ that is only available in source $j$, and a *hidden part* $c_{\mathrm{hidd}}^j$ that is available in the other data source but not in $j$, i.e.,

$c^j = \left[ c^j_{\text{shar}}, c^j_{\text{uniq}}, c^j_{\text{hidd}} \right]$. The shared part denotes variables that are available in both data sets and coincide for both data sources, $c^1_{\text{shar}} \triangleq c^2_{\text{shar}} \in \mathbb{R}^{q_{\text{shar}}}$. However, due to distribution shift, the unconditional distribution may differ between data sources, $p(c^1_{\text{shar}}) \neq p(c^2_{\text{shar}})$. A typical example may be the age of each individual: different studies often collect data in different age cohorts. The remaining variables are unique to each data source and may have different dimensionality, $q^j_{\text{uniq}}$. These could be specific variables, such as different brain volumes, and cataracts (a disease), which are not considered in each data source but related to the shared variable. With only two data sources, $c^1_{\text{uniq}} \triangleq c^2_{\text{hidd}}$ and vice versa.

We assume that the *conditional* distribution does not shift between data sets, that is, we assume that $p(c^1_{\text{hidd}}|c^1_{\text{shar}} = \xi) = p(c^2_{\text{uniq}}|c^2_{\text{shar}} = \xi)$ for any $\xi$. This assumption allows us to model the distribution between the different latent variables. Hence, we fit *stochastic* models $f^j$ that can sample from the conditional distributions, respectively: on source 1, we fit $f^1(c^1_{\text{shar}}) = \hat{c}^1_{\text{uniq}}$, which can be used in source 2 to impute the missing $\hat{c}^2_{\text{hidd}}$, and vice versa. Different parametric and non-parametric statistical methodologies can be employed for such a distribution estimation, such as structural equation modeling (Pearl, 2013; Pawlowski et al., 2020), Bayesian methods (Saatchi and Wilson, 2017), or GAN-based methods (van Breugel et al., 2021). In this work, we use Maximum Likelihood Estimation (MLE) of a hand-designed parametric model for all conditional and unconditional distributions, see Section D. For all latent factors, we fit models from which we can sample, instead of only predicting the most likely outcome.

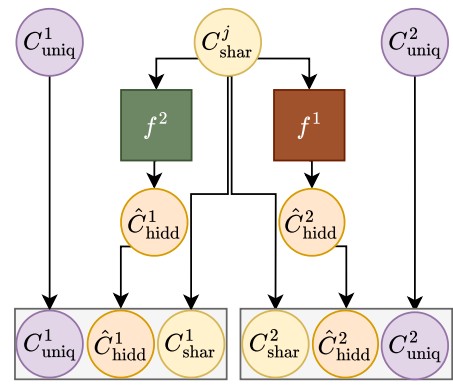

**Figure 2:** Latent space model of MSSG. $f^1$ approximates $c^2_{\text{hidd}}$ from the shared covariate $c^j_{\text{shar}}$ and vice versa. They are integrated into a joint latent space $\hat{c}^j = (c^j_{\text{shar}}, \hat{c}^j_{\text{hidd}}, c^j_{\text{uniq}})$. The generator $G$ gets the concatenated vector from a latent noise $z$ and the joint latent space $\hat{c}^j$.

### 2.3. Training Paradigm

Before training the MSSG, we first fit the latent space models $f^1, f^2$ as described in the previous section. In this work, all conditional models are parametrized by (generalized) linear models or conditional Gaussian Mixture Models (GMMs). We adapt the standard GAN training paradigm with alternating discriminator and generator update steps. Our proposed model has a StyleGAN backbone, but in principle, it can be used with most GANs. Figure 2 shows how the concatenated vector $(z, c^j_{\text{shar}}, \hat{c}^j_{\text{hidd}}.c^j_{\text{uniq}})$ is passed to the generator $G$ to generate images.

During the **generator step**, we sample the available ground-truth covariates $c^j_{\text{shar}}$ and $c^j_{\text{uniq}}$ directly from the training data sets, for both data sources at equal proportions. We then estimate the hidden variables from the latent space models fit in the initialization stage, $\hat{c}^1_{\text{hidd}} = f^2(c^1_{\text{shar}})$ and vice versa. We gather the latent data of both sources in a joint batch and sample $z$ from the noise distribution. We use $c$ and $z$ to generate an image with the generator, $\text{Img} = G(z, c_{\text{shar}}, c_{\text{uniq}}, \hat{c}_{\text{hidd}})$. We then compute predictions from the

discriminator, $\hat{y}, \hat{c}_{\text{shar}}, \hat{c}_{\text{uniq}}, \hat{c}_{\text{hidd}} = D(\text{Img})$, where $\hat{y}$ is the binary label if the image is generated or real, and propagate the loss as described in Section 2.1 to update the generator.

During the **discriminator step**, we generate covariates from the latent-variable model described in the previous Section 2.2. The shared and unique covariates are sampled separately for each data source. We sample the hidden conditional covariates from the corresponding latent space model from the other source. We then generate synthetic images with the generator and draw real samples from both sources at equal proportions. The discriminator is again trained to both distinguish between fake and real images and to predict the correct latent covariates from the images. However, we only compute the loss over the covariates for the real images to prevent shortcut learning, where the discriminator and generator cooperate to minimize the loss without solving the training task.

### 2.4. Inference Stage

After training the proposed MSSG, we can generate images while controlling all covariates from different sources jointly. To draw a fully random synthetic sample from any of the available sources, we can draw a sample from the shared covariate model, $c_{\text{shar}}^j$ and push it through latent space models $f^1, f^2$, to generate $c_{\text{uniq}}^j$ and $c_{\text{hidd}}^j$. Alternatively, of course, we can directly set the variables $c_{\text{shar}}$, $c_{\text{uniq}}$ and $c_{\text{hidd}}$ to any desired values. This is shown in Figure 2, where we describe how to use the joint latent covariables with GANs.

## 3. Experimental Evaluation

We first validate that MSSG can properly learn from multiple sources in the toy data set MorphoMNIST that allows full control of latent factors. Subsequently, we apply our method to real-world datasets – brain magnetic resonance images and retinal fundus images—from the UK Biobank and simulate a multi-source scenario with a data split. Lastly, we explore a fully realistic multi-source setting by incorporating images from ADNI into the UKB setting. We compare our MSSG model with single-source StyleGANs, which are trained without latent space models and only on a single source.

**Evaluation Metrics**  Various image quality metrics have been proposed, including the Inception Score (Salimans et al., 2016), Precision-Recall (Kynkäänniemi et al., 2019), Fréchet Inception Distance (FID) (Heusel et al., 2017), and Kernel Inception Distance (KID) (Gretton et al., 2006). Here, we focus on the FID and report KID and further metrics, namely Learned Perceptual Image Patch Similarity (LPIPS) (Zhang et al., 2018), Structural Similarity Index Measure (SSIM), Peak Signal-To-Noise Ratio (PSNR), in Section E.

Additionally, we evaluate the controllability of the latent factors in the generated images. Previous works such as the Intra-FID (Miyato and Koyama, 2018) do not apply to images with continuous labels and do not address the controllability of specific covariates. We propose a new metric, the *strata prediction score*, to evaluate the controllability of continuous covariates. We stratify test set samples into $m$ marginal bins per covariate, with each marginal bin containing 33% of the total sample size to maintain adequate representation. Within each bin, we generate $15,000$ images corresponding to the test set labels. Predictions for all covariates are made using separately trained prediction models on both real

and generated images. The strata prediction score is then calculated as the weighted average Pearson correlation coefficient between predicted covariates from generated and test set images across all bins. See Section C for details.

### 3.1. Validation in MorphoMNIST

**Setting** MorphoMNIST (Castro et al., 2019) is a semi-synthetic dataset derived from the MNIST benchmark. In contrast to MNIST, MorphoMNIST allows explicit control of morphological features like `thickness`, pixel `intensity`, `slant` (rotation) in addition to the `digit`.

**Table 1:** FID ↓ in three semi-multisource scenarios

| FID ↓ | Data sets | | |
| Methods | MorphoMNIST | MRI | Retina |
| --- | --- | --- | --- |
| Source 1 (baseline) | 3.28 | 22.11 | 30.19 |
| Source 2 (baseline) | 3.08 | 8.91 | 14.21 |
| Multi-source (half) | 3.13 | 13.81 | 14.62 |
| Multi-source (full) | **2.24** | **8.62** | **10.17** |

We modified the synthetic data generation model proposed by Pawlowski et al. (2020); Section D.1 provides detailed information. The first data source includes `thickness`, `digit`, and `intensity`, while the second source includes `thickness`, `digit`, and `slant`, i.e., the shared variables are $c_{\text{shar}} = [\texttt{thickness}, \texttt{digit}]$, with $c^1_{\text{uniq}} = c^2_{\text{hidd}} = \texttt{intensity}$ and $c^2_{\text{uniq}} = c^1_{\text{hidd}} = \texttt{slant}$.

**Evaluation** As a baseline, we train conditional StyleGANs on each source independently, each with a sample size of $N = 24,000$. Our multi-source StyleGAN can integrate data from different sources and thus can utilize more data than the single-source models. To disentangle the effect of increased sample size and covariate aggregation, we first train our model on a reduced data set with 12,000 samples from each data source (i.e., $N = 24,000$ in total), denoted by "half." We also train a model on the full data set of $N = 48,000$ images, denoted by "full."

Table 1 (first column) shows that at comparable sample size, our multi-source model achieves comparable image quality performance to the baseline methods while using both full data sets (which is not possible for the single-source models) leads to a considerable improvement in image quality. In Figure 3 we explore the controllability of the different models. As expected, our multi-source models can reliably control all three covariates, with slight improvements for larger sample sizes. The single-source baseline models can only control the respectively available covariates well but show low-to-moderate correlation with the unavailable covariates due to the correlation between thickness, intensity, and slant in the training data

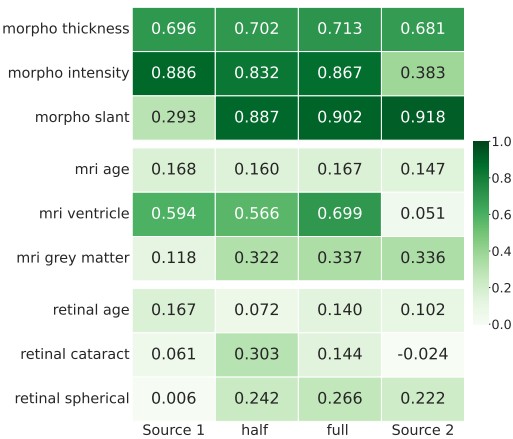

**Figure 3:** The strata weighted Pearson's correlation ↑ between the outputs of prediction models.

(see Section D.1). Qualitatively, in Figure 7 we fix two covariates and show that our model can control the remaining two variables; note that our model can jointly control the unique

variables `slant` and `intensity`, even without instances in the training data with both variables concurrently available.

### 3.2. Validation on Real-World Data

Next, we validate MSSG in real-world scenarios with more subtle data dependencies, focusing on two medical imaging modalities: brain MRI, used in diagnosing conditions like Alzheimer's Disease; and retinal fundus images, used in diagnosing ophthalmological conditions such as cataracts, and glaucoma, or diabetic retinopathy.

In the real-world setting, we use data from the UKB resource. To validate the models' controllability, we divide each dataset into two artificial sources, allowing for a comprehensive evaluation. This approach addresses the absence of covariate labels in real-world multi-source settings, which we further explore in the next section.

**Brain MRI**   We use the coronal middle slice of T1-weighted brain MRI scans, aligning them with the MNI atlas. We split the data set into two sources with $N = 13,414$ and $N = 13,408$ samples, respectively. The first source has `age` and `ventricle volume` as covariates, which are positively correlated due to ventricle enlargement with age (Kwon et al., 2014; Nestor et al., 2008). The second source has `age` and `grey matter volume`, which are negatively correlated (Giorgio et al., 2010; Callaert et al., 2014; Ramanoël et al., 2018).

Image quality between Source 1 and Source 2 differs strongly; our half-sized multi-source model achieves better performance than the average of the two models, and the full model outperforms all of the former models (Table 1, second column). As shown in Figure 3, all models only reach low-to-moderate (but comparable) controllability of the individuals' age. As in the MorphoMNIST example, ventricle and grey matter volumes can be jointly controlled moderately well with our multi-source model. Note that all three variables – `age`, `ventricle volume`, `grey matter volume` – are all very hard to discern from only single-slice MRI (Ballester et al., 2021), explaining the lower controllability performance compared to MorphoMNIST.

Figure 1 demonstrates the joint control of grey matter and ventricle volumes for fixed age. Independent control of ventricle volumes leads to their enlargement, while the grey matter consistently expands across every row, where ventricle and age remain constant. A subtle brightening effect at the periphery of the MR images aligns with the anatomical placement of grey matter at the brain's edge. An intriguing observation is the inverse relationship between ventricle volumes and grey matter volumes, suggesting a neurologically negative correlation.

**Retinal fundus images**   We again use `age` as a shared covariate. The first source includes `cataract` as a binary conditional covariate, the second source contains `spherical power` as a unique variable, representing the lens power required to correct myopia (nearsightedness) or hyperopia (farsightedness). Due to the low incidence of cataract cases in the UKB dataset ($\approx 3.9\%$) and to prevent class imbalances, we split the data into two sources, with 60% normal and 40% cataract images. Each source has approximately 1920 images in the training set. Due to the low sample size, in both sources, we included mirrored images and ADA-style data augmentation (Karras et al., 2020a).

In Table 1 image quality for the single-source models varies and the half-sized multi-source model is comparable to the better of the two, while the full multi-source model

outperforms all other models. Controllability is less stable than for the previous data sets (Figure 3). Spherical power can be controlled moderately well for those models that include it, but both age and cataract vary more strongly. Potentially, our "half" model overspecializes on modeling the cataract phenotype but underperforms on age, while the full model strikes a more balanced performance. Figure 14 again shows that MSSG can control the variables (cataract and spherical power) jointly when the age is set to 59.

### 3.3. Application in true multi-source setting

Finally, we investigate a real-world multi-source setting. We incorporate ADNI (Petersen et al., 2010), a clinical dementia dataset, with UKB (Section 3.2). From ADNI, we select covariates `age`, `left hippocampus`, and `right hippocampus` from SynthSeg (Billot et al., 2023). Studies underscore the significant correlation between hippocampal volumes and cognitive function (O'Shea et al., 2016; Evans et al., 2018), highlighting the relevance of these covariates. For the UKB cohort, we consider `age`, `ventricle`, and `grey matter` volumes. Notably, the ADNI cohort is older compared to UKB ($\approx 74$ versus $\approx 55$, respectively).

Table 2 compares FID scores; MSSG outperforms the other models on the joint data set, while the more specialized single-source models model their specific data sources more closely (which would be expected). Table 6 demonstrates that the multi-source model can model all respective covariates with simi-

**Table 2:** FID ↓ of UKB and ADNI in true multi-source setting. "Joint" denotes both cohorts are merged.

| FID ↓ Data sets | Source UKB | Methods Source ADNI | Multi-Source |
|---|---|---|---|
| UKB | 8.0 | 74.5 | 15.5 |
| ADNI | 66.1 | 16.8 | 34.3 |
| Joint | 26.6 | 21.83 | 19.3 |

lar performance as the specialized baseline models. In Figure 10, we show brain MRI generated by the multi-source model jointly trained on UKB and ADNI, and Section F.2 shows more qualitative examples of generated images.

## 4. Discussion & Conclusion

We introduced Multi-source StyleGAN (MSSG), a conditional generative image model capable of learning from multiple disparate data sources concurrently. Our experiments demonstrate that integrating multi-source data does not compromise image quality compared to single-source generation, and it can enhance data quality by leveraging a larger dataset. Through various case studies, we validated MSSG's ability to control variables from different sources collectively, even without access to paired variables. We believe MSSG can address data scarcity and label scarcity issues in medical image data, especially for rare diseases. However, a limitation of our current method is its reliance on a hand-designed latent space model, since we wanted to ascertain that the latent model has a good fit onto the true latent distribution. Future work could explore using non-parametric general-purpose models like DECAF (van Breugel et al., 2021) as a drop-in replacement. Additionally, extending our multi-source integration approach to other generative models such as VAEs and diffusion models holds promise for future research.

## Acknowledgments

The authors thank the participants of the UK Biobank study and the ADNI study. This research has been conducted using the UK Biobank Resource under Application Number 77717. Furthermore, data used in preparation of this article were obtained from the Alzheimer's Disease Neuroimaging Initiative (ADNI) database (adni.loni.usc.edu). As such, the investigators within the ADNI contributed to the design and implementation of ADNI and/or provided data but did not participate in the analysis or writing of this report. A complete listing of ADNI investigators can be found at: http://adni.loni.usc.edu/wp-content/uploads/how_to_apply/ADNI_Acknowledgement_List.pdf

## Data availability

In this work, three types of datasets are used. The Morpho-MNIST dataset is available in the Github repository (https://github.com/dccastro/Morpho-MNIST). The MR imaging is available by applications from the UK Biobank (https://www.ukbiobank.ac.uk/register-apply) and the ADNI database (https://adni.loni.usc.edu/data-samples/access-data/). Last but not least, the retinal images are available by the application from the UK Biobank (https://www.ukbiobank.ac.uk/register-apply).

## Funding

This work was kindly supported by the German Ministry of Research and Education (Bundesministerium für Bildung und Forschung - BMBF) within the project 'Syreal' (Grant No. 01/S21069A). The funders had no role in the study design, data analysis, interpretation, or report writing.

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

## Appendix A. Background

GANs are among state-of-the-art architectures in DL to generate high-dimensional data (Wang et al., 2020). The backbone of a GAN method contains two networks, a generator network $G$ and a discriminator network $D$ (Goodfellow et al., 2014). The generator produces synthetic data from a noise vector $z$ and the discriminator distinguishes it from real data. In recent years, frameworks such as StyleGAN (Karras et al., 2019, 2020b, 2021) and BigGAN (Brock et al., 2019) have developed GANs to generate high-quality images at high resolutions. Recent diffusion models (Sohl-Dickstein et al., 2015) have been shown to achieve promising image synthesis (Rombach et al., 2021; Song et al., 2020, 2021; Ho et al., 2020), but have slower sampling times than GANs (Dhariwal and Nichol, 2021), require larger training data sets (Moon et al., 2022), and can have a slow, computationally expensive training process (for example, the training time on the CelebA-64x64 dataset required around 24 hours with 16 V100 GPUs (Wang et al., 2023)).

**Conditional GANs** Conditional GANs (Mirza and Osindero, 2014) were introduced to generate image data conditional on categorical factors. Several studies modified and improved the performance of conditional GANs (Miyato and Koyama, 2018; Brock et al., 2019; Karras et al., 2020b, 2021). Most works only handle *categorical* features instead of continuous or mixed-type labels and usually ignore dependencies between input labels. One alternative is ccGANs (Ding et al., 2021, 2023); however, ccGANs only tackle univariate continuous labels, such as angles and ages. Our proposed method targets multiple continuous conditional labels and the problem of missing covariates in a database.

**Multi-Source/Multi-Domain Image Generation** There have been several prior works on integrating multiple sources into a joint generative model but with a different focus from our work. Most prominent are works on image-to-image translation; e.g., CycleGAN (Zhu et al., 2017) is trained with cycle-consistency loss using unpaired images to translate images between two domains. DVG-Face (Fu et al., 2021) focuses on generating dual heterogeneous paired face images to preserve identity. Kang et al. (2023) proposed a method that translates medical images across domains while preserving structural information during translation. This method requires an image from the target domain and the structure and texture features from the source domain as input. Furthermore, StarGAN v2 (Choi et al., 2020) translates images between domains with a single generator and takes an image and a style code as input to increase the diversity of translated images in the target domain. MPG (Han et al., 2021) proposed a multi-attribute version of StyleGAN2 to generate pizza images with specified ingredients and views. In this approach, a pre-trained view attribute regressor is used to impute the missing values of labels. Our work differs from these approaches, as our proposed method trains on two datasets simultaneously and uses latent space models to impute missing labels.

In another related research problem, image harmonization techniques (Bashyam et al., 2022; Liu et al., 2021; Beizaee et al., 2023) address domain transfer, which is caused, e.g., by different MRI scanners or data collection setups (Liu et al., 2021). Here, techniques take images as input and transfer them to different domains, but they are not able to control the image characteristics, such as ventricle volumes.

## Appendix B. Multi-Source StyleGAN

**Pseudo Code for the training paradigm**    Here, we provide the pseudo-code for training the proposed multi-source GAN method Algorithm 1, which is described in Section 2.3. For simplicity, we will focus on the case of two sources, $D^1$ and $D^2$. The required latent space models $f^1$ and $f^2$ for the GAN training paradigm are hand-designed parametric models as described in Section 2.2, but they are integrated into the training script.

---

**Algorithm 1:** Training a Multi-Source GAN

---

**Require:** $f^1, f^2, G, D$
**Input:** $D^1 = \left\{(x_i^1, c_i^1)\right\}_i, D^2 = \left\{(x_i^2, c_i^2)\right\}_i$

**1** Initialize Generator $G$ and Discriminator $D$ with random weights
**2** Set number of steps $S$ and batch size $B$
**3** Set learning rates $\eta_G$ and $\eta_D$
**4** **for** $step \leftarrow 1$ $to$ $S$ **do**
**5**   Sample batches of real images $\{x_1^1, \ldots, x_B^1\}, \{x_1^2, \ldots, x_B^2\}$ from datasets $D^1$ and $D^2$
**6**   Sample batches of real labels $\{c_1^1, \ldots, c_B^1\}, \{c_1^2, \ldots, c_B^2\}$ from datasets $D^1$ and $D^2$
**7**   Estimate the hidden variables $\hat{c}_{\text{hidd}}^1 = f^2(c_{\text{shar}}^1), \hat{c}_{\text{hidd}}^2 = f^1(c_{\text{shar}}^2)$ by using latent space models $f^1$ and $f^2$
**8**   Sample batches of noise vectors $\{z_1^1, \ldots, z_B^1\}, \{z_1^2, \ldots, z_B^2\}$ from the noise distribution
**9**   Generate fake images $\{\text{Img}_1^1 = G(z_1^1, c_{1,\text{shar}}^1, c_{1,\text{uniq}}^1, \hat{c}_{1,\text{hidd}}^1), \ldots, \text{Img}_B^1 = G(z_B^1, c_{B,\text{shar}}^1, c_{B,\text{uniq}}^1, \hat{c}_{B,\text{hidd}}^1)\}$ using Generator $G$ for $D^1$
**10**   Generate fake images $\{\text{Img}_1^2 = G(z_1^2, c_{1,\text{shar}}^2, c_{1,\text{hidd}}^2, \hat{c}_{1,\text{uniq}}^2), \ldots, \text{Img}_B^2 = G(z_B^2, c_{B,\text{shar}}^2, c_{B,\text{hidd}}^2, \hat{c}_{B,\text{uniq}}^2)\}$ using Generator $G$ for $D^2$
**11**   Concatenate images $\{\text{Img}_1^1, \ldots, \text{Img}_B^1, \text{Img}_1^2, \ldots, \text{Img}_B^2\}$ and labels $\{\hat{c}_1^1, \ldots, \hat{c}_B^1, \hat{c}_1^2, \ldots, \hat{c}_B^2\}$ from two sources
**12**   Use data from the specific loss of each covariate to update the Discriminator
**13**   Use data from the specific loss of each covariate (only with generated images) to update the Generator
**14** **end**

---

**Generalization to more than 2 data sources**    Our proposed multi-source GAN can be generalized to $\geq 3$ data sources. Depending on which latent variables are available in which data source, this leads to modeling choices within the latent space models. For example, if a variable is available in sources 1 and 2 but not in source 3, we have to decide if we want to sample it from a model derived from source 1 or source 2, or perhaps alternate between these two. Therefore, this will be a more application-specific question. In this paper, we mainly focus on dealing with two sources.

## Appendix C. Implementation Details

**Training details**    We implemented MSSG, using the StyleGAN3 source code (Karras et al., 2021), keeping many of the default parameters. The generator in MSSG is based on the StyleGAN3 generator, employing a latent noise vector $z \in \mathbb{R}^{512}$ and a conditional

latent vector integrated with our joint conditional vector. The mapping network of the generator consists of two fully connected layers, producing another latent vector $w \in \mathbb{R}^{512}$. The discriminator, a StyleGAN2 Discriminator, is set to default configurations. Both the generator and discriminator are trained using Adam optimizers. Models were trained with ADA data augmentation to prevent overfitting. X-Flip was applied to the retinal data sets, as the data sets are small. We adapted the conditional loss function from StyleGAN3, modifying the discriminator's output to predict correct labels and disabling the mapping network in the discriminator. In preliminary experiments, we found that weighing the fake/real loss and the covariate prediction loss equally led to considerably decreased image quality. Instead, we multiply the covariate losses by a scaling parameter $\lambda$, set to 0.1 throughout our real-world experiments. For the first few iterations of training, we grow $\lambda$ exponentially from 0 to its target value. This helps emphasize image quality at the start, reaching a maximum of 0.9 in semi-multisource cases and 0.1 in the true multi-source case.

With the extension to the real multi-source data sets, we can express also the property of sources as an extra covariate $c_{\text{source}}^j$ in our conditional vector $c^j$. This allows us to explicitly sample from either of the data sources $c_{\text{source}}^j$ instead of it being expressed by the latent noise vector $z$.

We trained our models on 2 A100 GPUs until the convergence of the FID score on a validation set. The duration of training is data-set dependent. For MorphoMNIST, the convergences of the FID score took 8192 steps for the proposed "half" model and 22937 steps for the proposed "full" model, as the resolution of images is $32 \times 32$, and the baseline models took 19661 and 18022 steps respectively. For MRI and retinal fundus image experiments, we used a resolution of $128 \times 128$. The convergence of the training took 13107 steps for the proposed "half" model and 18022 for the proposed "full" model on the MRI experiments. The baseline models on single-source data sets took 29286 and 19661 steps to reach the convergence. In the retinal fundus image experiments, the proposed "half" model took 57384 steps to converge and the proposed "full" model took 47937 steps to converge. The baseline single-source models took 39601 and 30719 steps to converge.

**Evaluation details** Quantitative evaluation consists of image quality assessment and controllability analysis. Image quality is evaluated using FID and KID, while controllability is measured through the proposed strata prediction score. In this evaluation, test sets are stratified for each covariate, divided into $m = 3$ marginal bins. Each marginal bin contains 33% of the total samples. With three covariates in our experiments, this results in $3^3 = 27$ subsets. Regression models, specifically, ResNet50s, are employed in each stratum to predict covariates from both test sets and generated images. A total of $15,000$ images are generated for each stratum and the score is computed as the weighted (by stratum size) average of the Pearson correlation coefficients of predicted outputs from generated and test set images.

For MorphoMNIST, the ResNet50 regression models yield high performance, with Pearson correlation coefficients of 0.978 for `thickness`, 0.996 for `intensity`, and 0.999 for `slant`.

On the real-world MRI data from the UKB cohort, ResNet50 regression models achieve correlations of 0.78 for age, 0.97 for ventricle volumes, and 0.87 for grey matter volumes. In the ADNI cohort, ResNet50 models for age, left hippocampus, and right hippocampus achieve correlations of 0.964, 0.980, and 0.984 respectively.

For retinal fundus images in the UKB cohort, ResNet50 models exhibit correlations of 0.863 for age and 0.910 for spherical power. The binary covariate cataract achieves an accuracy of 0.70 and a Pearson correlation coefficient of 0.338 (we use Pearson correlation to stay consistent with the other measures).

Prediction models were trained using ResNet18, ResNet50, and ResNet100, and the model with the best performance was selected for our metric.

## Appendix D. Parametric Models

In the example of training two sources, our proposed method, MSSG, requires latent models $f^2$ to approximate the hidden covariates $c_{\text{hidd}}^j$ for the first source $D^1$, and vice versa. In this work, we designed latent space models by using parametric models with MLE. That means that, e.g., in the case of a linear regression model, we also need to estimate the standard deviation of the noise term in addition to the weights and bias. For example, assume that the shared variable is age, the unique variable in source 1 is blood_pressure, and the unique variable in source 2 is sex. We may then fit a linear regression model blood_pressure $= f^1(\text{age}) = \alpha\text{age} + \beta + \epsilon$ with $\epsilon \sim \mathcal{N}(0, \sigma^2)$ on data source 1 to estimate $\alpha$, $\beta$, and $\sigma$. At the same time, we fit a logistic regression to model $\mathbb{E}[\text{sex}] = \text{sigmoid}(\gamma\text{age} + \delta)$ on data source 2 for $\gamma$ and $\delta$ (i.e., $f^2(\text{age})$ is 1 with probability $\text{sigmoid}(\gamma\text{age} + \delta)$ and 0 otherwise). Given these models, for an instance from source 1 we can then e.g. randomly sample from $p(\text{sex}|\text{age})$ given the logistic regression model $f^2(\text{age})$.

### D.1. Synthetic Data

**Morphological digits** We synthesize the MNIST dataset with the tool from Castro et al. (2019), randomly split it into two data sources, and train on them with our proposed method. The first data source has access to thickness, intensity, and digit. The second data source has access to thickness, slant, and digit. We modified the data generation process from Pawlowski et al. (2020) to generate synthetic morphological digits. Thickness is sampled from a gamma distribution $\Gamma$. Given thickness, intensity, and slant are sampled as follows:

$$\text{thickness} = 0.5 + \epsilon_t, \qquad\qquad \epsilon_t \sim \Gamma(10, 5)$$
$$\text{intensity} = 191 \cdot \sigma(0.5 \cdot \epsilon_I + 2 \cdot \text{thickness} - 5) + 64, \qquad\qquad \epsilon_I \sim \mathcal{N}(0, 1)$$
$$\text{slant} = 56 \cdot \tanh(0.3 \cdot \epsilon_S + \text{thickness} - 2.5), \qquad\qquad \epsilon_S \sim \mathcal{N}(0, 1)$$

$\sigma(\cdot)$ is the sigmoid function. The value range of the intensity is therefore between $[64, 255]$ and the digits rotate in the range of $[-56, 56]$.

To estimate the covariates in our latent model, for thickness, we use a beta distribution thickness $\sim \beta(4.13, 9.84)$. Intensity and slant are estimated by conditioning on thickness, $p(\text{intensity}|\text{thickness})$ and $p(\text{slant}|\text{thickness})$. We fitted the known covariates into a non-linear least square function with the sigmoid function to optimize the parameters. Figure 4 shows that our latent models can approximate the actual label distributions well in both sources.

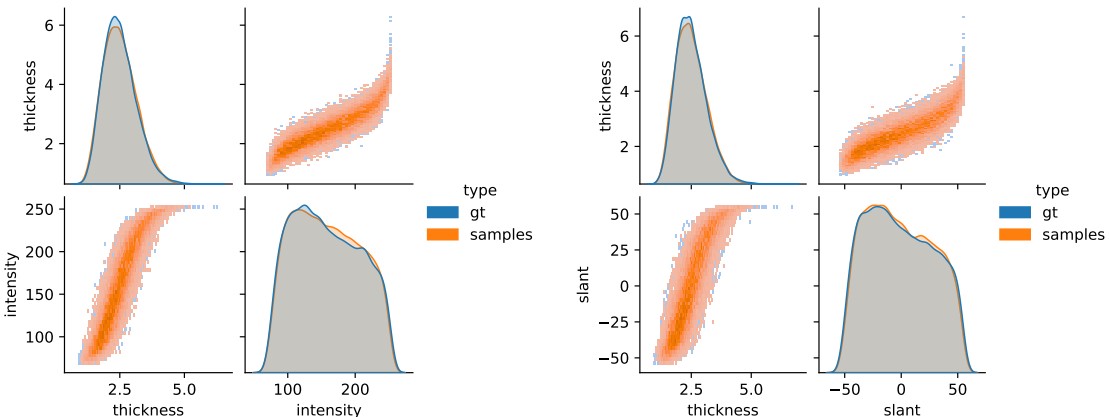

($a$) Latent model $f^1$ samples intensity based on the sampled thickness.

($b$) Latent model $f^2$ samples slant based on the sampled thickness.

**Figure 4:** Samples from latent models fit the ground-truth MorphoMNIST distributions. The blue legend shows samples from the ground truth and the orange dots show samples from latent models.

## D.2. Real-World Data

**Brain MRI** For model training, covariates are fitted into separate latent space models. We show here also how latent models $f^j$ approximate the covariates $c^j$. For the UK Biobank, Figure 5($a$) shows that ventricle volumes and grey matter volumes are estimated well by the latent model with the given ages. `age` is rescaled to the interval $[0, 1]$ and modeled by a beta distribution, while `ventricle` and `grey matter volumes`, conditioned on `age`, can be sampled together by a Gaussian mixture regression model with 10 Gaussian components.

In the ADNI data set, we have three covariates, `age`, `left hippocampus`, and `right hippocampus`. Another 8-component GMM regression is applied to learn to sample the left hippocampus and right hippocampus. Age is again modeled by a beta distribution (with independent parameters) and the left and right hippocampus are predicted by age.
Figure 5($b$) depicts the fit of the trained latent model for the left and right hippocampus, conditioned on the age.

In Figure 6($a$) we illustrate how sampling from either of the two latent models leads to varying latent space distributions. Given that the UKB cohort exhibits a younger age distribution, the latent space model trained on the ADNI dataset shows a distribution shift when sampling hippocampal volumes based on the age distribution of the UKB.

**Retinal fundus images** The shared covariate `age` was modeled as a rescaled beta distribution, with minimum and maximum values set based on observed data. The binary variable `cataract` is incorporated into a logistic regression model conditioned on `age`. `Spherical power` is modeled using a Gaussian Mixture Model regression with 13 Gaussian components, also conditioned on `age`. Figure 6($b$) shows the distributions of the ground truth labels and samples from the latent model.

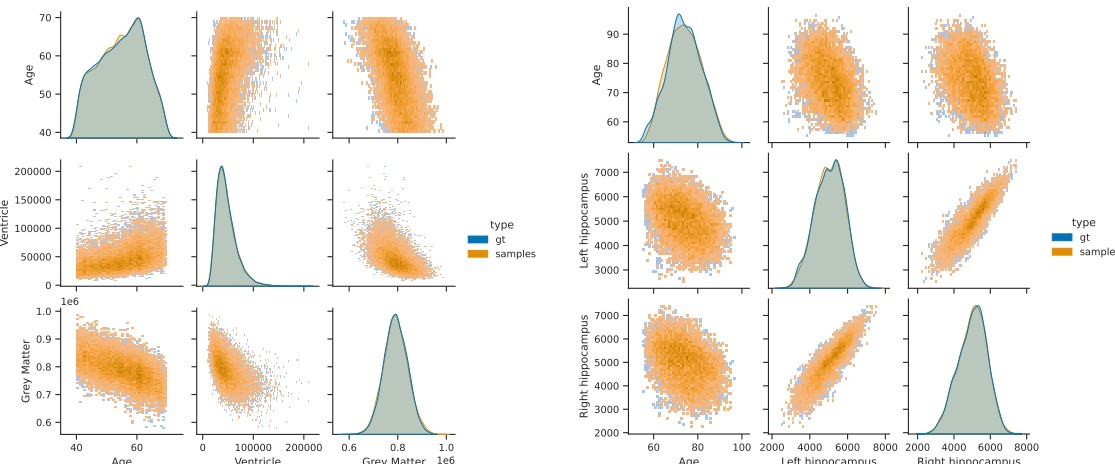

(*a*) The latent model samples ventricle volumes and grey matter volumes by conditioning on the given ages. Ventricle volumes increase with higher age, and grey matter volumes shrink with higher age.

(*b*) Left and right hippocampus volumes conditioned on the given ages. The hippocampus volumes shrink slightly with higher age.

**Figure 5:** Latent models sample the distributions of the UKB and ADNI cohorts in the MRI experiments.

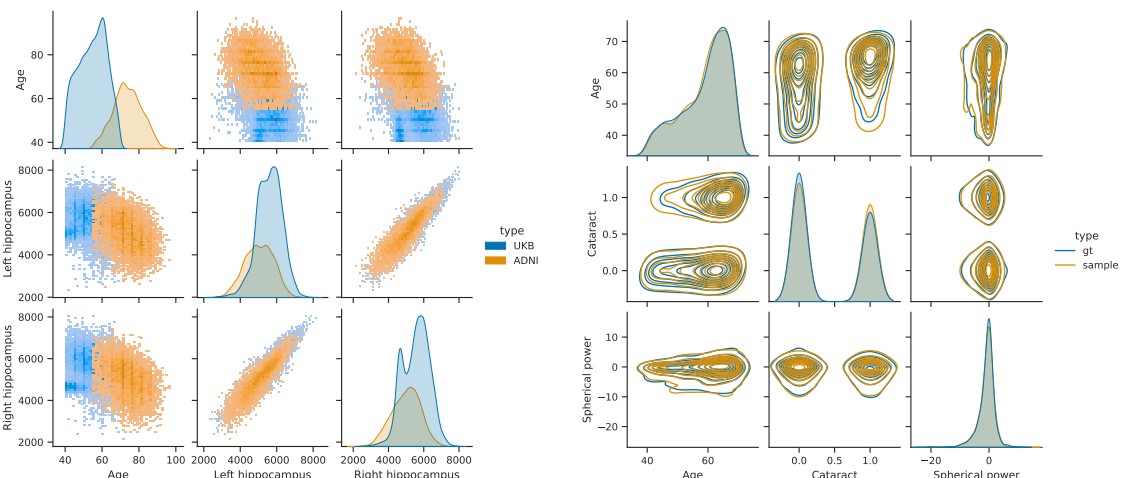

(*a*) Latent model trained on the ADNI cohort samples hippocampal volumes by conditioning on two different age distributions from the UKB and ADNI cohorts.

(*b*) Cataract and spherical power are sampled from the trained latent model conditioned on age.

**Figure 6:** Latent models in the MRI and retinal fundus images experiments.

## Appendix E.  Additional Experimental Results

In this section, we show additional training information and KID scores in Table 3 for semi-multisource scenarios and Table 5 for the real-world multisource MRI. Furthermore, the evaluations of pairwise metrics for semi-multisource scenarios, i. e. LPIPS, SSIM, and PSNR, are depicted in Table 4. Nevertheless, it is not possible to evaluate the real-world multisource MRI with pairwise metrics, since the ground truth covariates do not necessarily exist.

### E.1.  Validation in synthetic data

**MorphoMNIST**   The experimental results of KID in Table 3($a$) differ slightly from FID scores reported in Section 3.1. However, by boosting the data size, our proposed model reduces the KID score and reaches comparable results. The proposed method can control three covariates, whereas single-source models can only control the existing covariates in data sets. Furthermore, Table 4($a$) shows the results of pairwise metrics, i. e. LPIPS, SSIM, and PSNR. In these evaluations, we compared generated with real images from the test set, given the same covariates. The proposed method outperforms the single-source StyleGAN3 models. By doubling the data size, our proposed method performs slightly better in the evaluations of SSIM and PSNR.

**Table 3:** KID ↓ and additional information in three different semi-multisource scenarios. "GM" denotes grey matter volumes, "SP" denotes spherical powers.

($a$) MorphoMNIST

| Methods | KID ↓ mean (std) | Training samples | Covariates |
|---|---|---|---|
| Source 1 (baseline) | 0.00121 (0.00238) | 24000 | Thickness, Intensity |
| Source 2 (baseline) | 0.00087 (0.00208) | 24000 | Thickness, Slant |
| Proposed (half) | 0.00128 (0.00225) | 24000 | Thickness, Intensity, Slant |
| Proposed (full) | 0.00088 (0.00278) | 48000 | Thickness, Intensity, Slant |

($b$) MRI: UKB

| Methods | KID ↓ mean (std) | Training samples | Covariates |
|---|---|---|---|
| Source 1 (baseline) | 0.02681 (0.00418) | 13414 | Age, Ventricle |
| Source 2 (baseline) | 0.00818 (0.00231) | 13408 | Age, GM |
| Proposed (half) | 0.01632 (0.00260) | 13411 | Age, Ventricle, GM |
| Proposed (full) | 0.00932 (0.00230) | 26822 | Age, Ventricle, GM |

($c$) Retina: UKB

| Methods | KID ↓ mean (std) | Training samples | Covariates |
|---|---|---|---|
| Source 1 (baseline) | 0.02029 (0.00400) | 1922 | Age, Cataract |
| Source 2 (baseline) | 0.00811 (0.00182) | 1918 | Age, SP |
| Proposed (half) | 0.00926 (0.00201) | 1920 | Age, Cataract, SP |
| Proposed (full) | 0.00333 (0.00124) | 3840 | Age, Cataract, SP |

**Table 4:** Pairwise metrics: LPIPS ↓, SSIM ↑, and PSNR ↑ in three different semi-multisource scenarios.

(*a*) MorphoMNIST

| Methods | LPIPS ↓ mean (std) | SSIM ↑ mean (std) | PSNR ↑ mean (std) |
|---|---|---|---|
| Source 1 (baseline) | 0.093 (0.004) | 0.345 (0.013) | 34.7 (0.143) |
| Source 2 (baseline) | 0.122 (0.004) | 0.326 (0.013) | 34.8 (0.107) |
| Proposed (half) | 0.064 (0.004) | 0.468 (0.020) | 35.5 (0.121) |
| Proposed (full) | 0.065 (0.003) | 0.471 (0.017) | 35.6 (0.146) |

(*b*) MRI: UKB

| Methods | LPIPS ↓ mean (std) | SSIM ↑ mean (std) | PSNR ↑ mean (std) |
|---|---|---|---|
| Source 1 (baseline) | 0.102 (0.001) | 0.611 (0.003) | 31.5 (0.026) |
| Source 2 (baseline) | 0.094 (0.001) | 0.588 (0.003) | 31.3 (0.021) |
| Proposed (half) | 0.072 (0.001) | 0.617 (0.003) | 31.6 (0.035) |
| Proposed (full) | 0.071 (0.001) | 0.654 (0.003) | 31.6 (0.030) |

(*c*) Retina: UKB

| Methods | LPIPS ↓ mean (std) | SSIM ↑ mean (std) | PSNR ↑ mean (std) |
|---|---|---|---|
| Source 1 (baseline) | 0.347 (0.006) | 0.429 (0.008) | 28.8 (0.020) |
| Source 2 (baseline) | 0.221 (0.007) | 0.541 (0.011) | 28.9 (0.030) |
| Proposed (half) | 0.204 (0.012) | 0.572 (0.009) | 29.0 (0.031) |
| Proposed (full) | 0.193 (0.008) | 0.581 (0.010) | 29.0 (0.043) |

## E.2. Validation in real-world data sets

### E.2.1. MRI

**Synthetic multi-source MRI** Similar to the experimental results for MorphoMNIST, we provide results of KID in Table 3(*b*). The tendency of the results is similar to those of FID reported in Section 3.2. However, the source 2 model performs better than our proposed "full" model. The proposed "half" model performs comparably to the average of the source 1 and source 2 models. Table 4(*b*) depicts the results of pairwise metrics. Besides the evaluation of PSNR, in which the results are similar between the proposed method and the baselines, the multi-source StyleGAN3 outperforms the single-source baselines on the evaluations of LPIPS and SSIM.

**True multi-source MRI** Table 5 again shows the KID. Interestingly, our proposed MSSG outperforms the specialized single-source UKB model on the UKB test set. Furthermore, on the joint test sets (UKB and ADNI) our model still reaches the lowest KID score compared to the single-source models. Table 6 depicts the results of the controllability of models.

**Table 5:** KID mean (std) ↓ for each method in corresponding test sets in true multi-source MRI setting

| KID ↓ mean (std) | **Methods** | | |
| --- | --- | --- | --- |
| **Data sets** | Source UKB | Source ADNI | Multi-Source |
| **UKB** | 0.00771 (0.00247) | 0.08759 (0.00804) | 0.00660 (0.00167) |
| **ADNI** | 0.08897 (0.00956) | 0.01353 (0.00355) | 0.02674 (0.00438) |
| **Joint** | 0.02848 (0.00698) | 0.02182 (0.00525) | 0.01023 (0.00344) |

**Table 6:** CS: Correlation score ↑ on corresponding covariates in true multi-source setting. Vntr: Ventricle, GM: Grey Matter, LH: Left Hippocampus, RH: Right Hippocampus

| CS ↑ | **Methods** | | |
| --- | --- | --- | --- |
| **Covariates** | Source UKB | Source ADNI | Multi-Source |
| Age (UKB) | 0.704 | -0.163 | 0.509 |
| Age (ADNI) | 0.264 | 0.402 | 0.532 |
| Vntr | 0.958 | -0.601 | 0.879 |
| GM | 0.827 | -0.189 | 0.755 |
| LH | -0.316 | 0.865 | 0.696 |
| RH | -0.486 | 0.886 | 0.845 |

E.2.2. RETINAL FUNDUS IMAGES

**Synthetic multi-source Retina**  Table 3(c) illustrates the KID scores of the models alongside the number of training samples. Similar to the trends observed in Table 1 for FID scores, the KID scores demonstrate a consistent pattern. The proposed "half" model achieves a comparable score to the source 2 model and outperforms the average of the source 1 model and source 2. Upon increasing data samples, the "full" model significantly reduces the KID score. Additionally, despite being trained on a low-data-sample regime, all models, besides the source 1 model, demonstrate the capability to generate high-quality images. Table 4(c) shows similar results as from the MRI use case. Nevertheless, the results are generally worse than those on the MRI data. This could be caused by the availability of fewer data samples (< 2000 for each source). In general, our proposed multi-source models reach lower LPIPS scores and higher SSIM scores, compared with the single-source models. Nevertheless, the PSNR scores are comparable between multi-source and single-source models.

# Appendix F. Visualizing generated imaging from MSSGs

Here, we further explore the multi-factor manipulation capabilities visually.

## F.1. Synthetic Data: MorphoMNIST

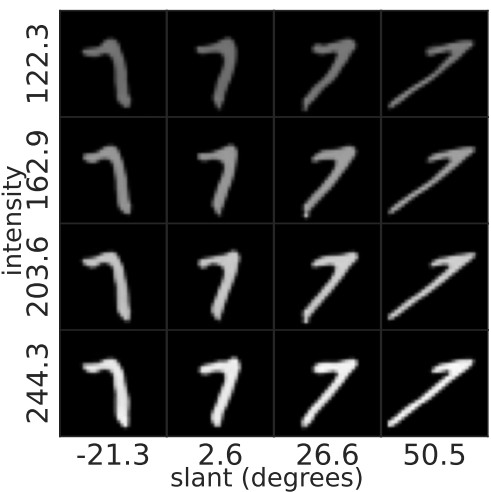 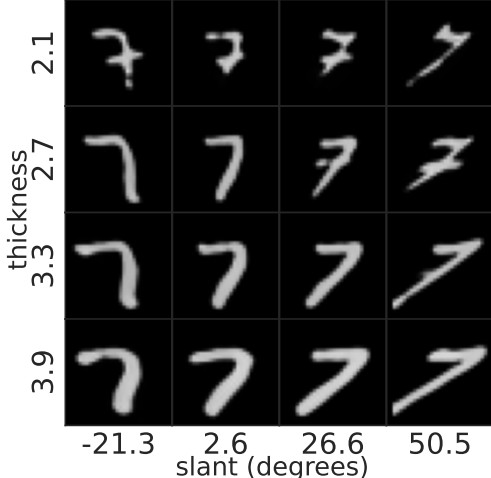

($a$) Varying `intensity` and `slant` with fixed `thickness` (2.99). Intensity and slant vary.

($b$) Varying `thickness` and `slant` with fixed `intensity` (183.289).

**Figure 7:** Morphological digits generated using the proposed "full" MSSG.

**Synthetic multi-source morphological digits** In this section, we present additional examples generated by the multi-source GAN. These figures were produced through the interpolation of covariate values, ranging from the minimal to the maximal values, specifically within a 30% range. This approach ensures the exclusion of outliers with extreme covariate values. As depicted in Figure 7 and Figure 8($a$), the multi-source GAN adeptly controls three continuous covariates—thickness, intensity, and slant. In Figure 7($a$), the shared covariate `thickness` is set to the mean value of the test set, which is 2.9. Intensity and slant are modified, with images exhibiting an increase in intensity column-wise, resulting in a brighter appearance. Additionally, there is a rotation effect from left to right row-wise, ranging from −21.3 degrees to 50.5 degrees. Conversely, Figure 7($b$) demonstrates various combinations of covariates (thickness and slant) with a fixed intensity. Here, the images become thicker column-wise, and there is a rotation effect from left to right within a single row. Finally, in Figure 8($a$), the slant (covariate from Source 2) is set to a constant value, while the generated images are controlled by thickness and intensity. In this scenario, the images become thicker in a column-wise manner while intensifying from left to right within each row.

### F.2. Real-World Data: MRI

**Synthetic multi-source MRI** We split UKB randomly into two data sources and trained them with our proposed method. Here, we show more generated examples from the multi-source GAN. We created the following figures by interpolating the values of covariates, starting from the minimal 30% until the maximal 30%. Figure 8($b$) and Figure 9 show the multi-source model can control three covariates in parallel. Figure 8($b$) shows the tendency

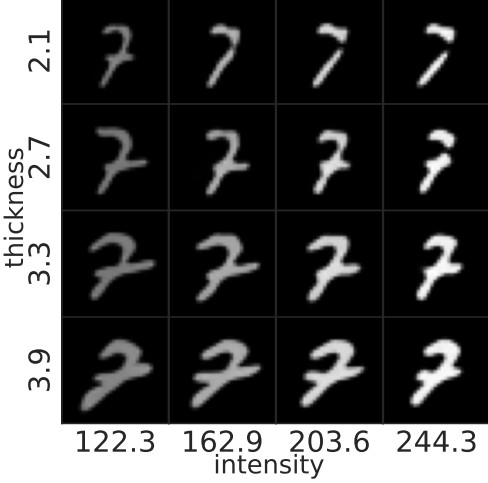

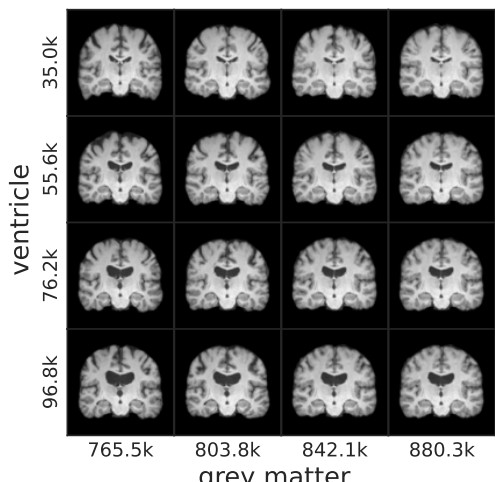

($a$) Morphological digits were generated using the proposed "full" multi-source GAN. Here, the covariate `slant` is set to 14.612 degree, and the model controls morphological attributes, the shared covariate (thickness), and intensity from Source 1 of digits.

($b$) Coronal MRIs were generated using the proposed "full" multi-source GAN. Another example on controlling ventricle and grey matter respectively when the age is fixed to 59.

**Figure 8:** Morphological digits and Coronal MRIs (two use cases) generated using the proposed "full" MSSG.

that if ventricle volumes increase row-wise, the grey matter volumes decrease (the slits on the edge of the brain becomes wider). On the other hand, if the grey matter volumes increase column-wise, this causes the shrinkage of ventricle volumes. This is related to the anatomical nature of these two covariates in Figure 5($a$). Ventricle volumes can be controlled monotonically well. In Figure 9($a$), the age and grey matter volumes are modified. The grey matter increases horizontally from left to right. However, the age-related changes are quite small. Figure 9($b$) shows the change in brains when the age and ventricle volumes are manipulated. Again, the size of ventricle volumes increases monotonically.

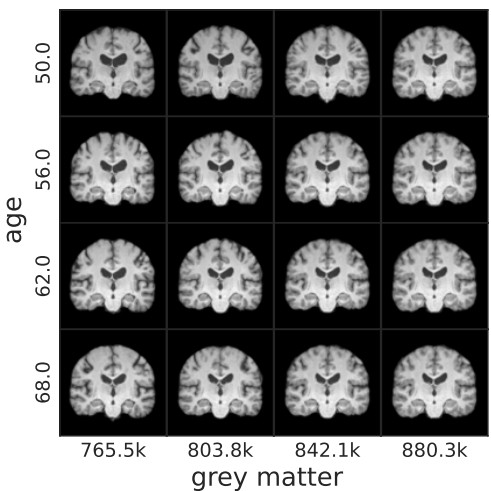
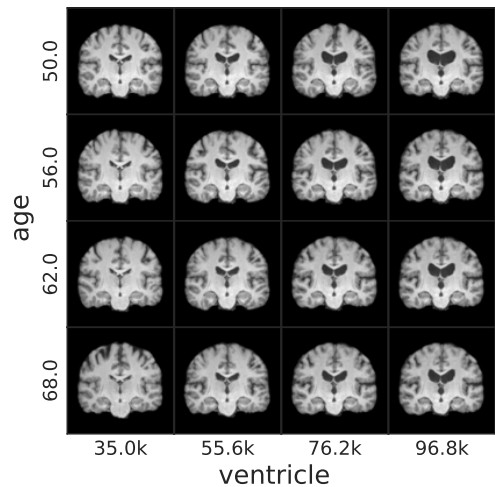

(*a*) Ventricle is set to a constant value (65876.6), MSSG generates images by controlling covariates (age and grey matter volumes).

(*b*) Grey matter is set to a constant (822924.9), MSSG generates images by controlling covariates (age and ventricle volumes).

**Figure 9:** Coronal MRI were generated using our "full" multi-source GAN.

**True multi-source MRI**    We also implemented our experiments on the true multi-source data sets, i.e. UKB and ADNI. Figure 10(a), demonstrates control over a younger age distribution from UKB while increasing volumes of the right hippocampus column-wise. Additionally, in Figure 10(b), the model regulates an older age distribution from ADNI, increasing grey matter volumes column-wise, with noticeable changes in brain volumes. Figure 11 shows additionally that the proposed MSSG can not only control covariates, i.e. ventricle volumes and left hippocampus, but also modify data sources. Figure 12 gives two further examples that MSSG controls various covariates across two data sources and generates reasonable images.

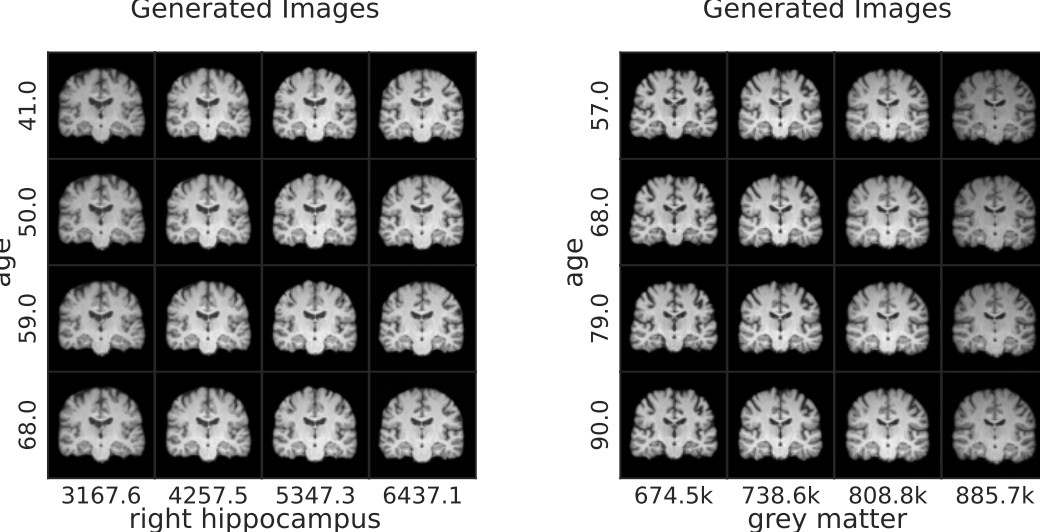

(a) MSSG controls the younger age distribution from UKB row-wise. In the meanwhile, the right hippocampus is modified column-wise.

(b) MSSG regulates the older age distribution from ADNI row-wise and increases grey matter volumes column-wise.

**Figure 10:** Generated MRI from the proposed MSSG trained on the UKB and ADNI.

Generated Images

Generated Images

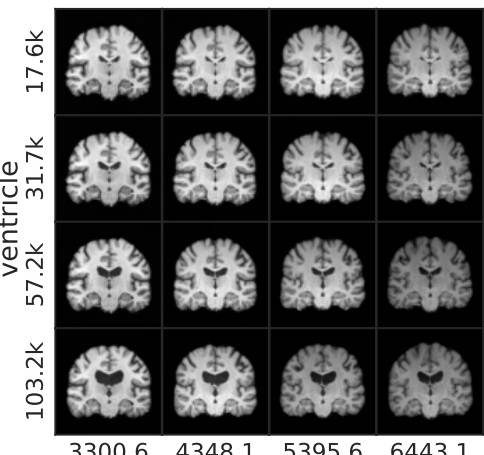

(*a*) MSSG generates images with specific ventricle sizes and left hippocampus sizes in the style of the UKB cohort.

(*b*) MSSG regulates ventricle volumes and left hippocampus volumes and generates images in the style of the ADNI cohort.

**Figure 11:** Generated MRI from the proposed MSSG in the true multi-source scenario by controlling ventricle volumes and left hippocampus volumes.

Generated Images

Generated Images

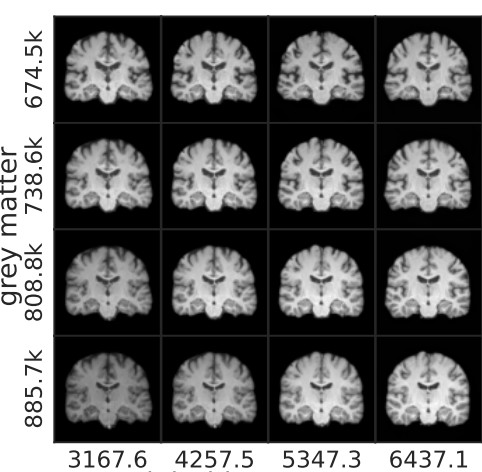

(*a*) MSSG generates images with specific ages and ventricle sizes in the style of the ADNI cohort.

(*b*) MSSG generates images with different grey matter volumes and right hippocampal volumes in the style of the UKB cohort.

**Figure 12:** Generated MRI from the proposed MSSG in the true multi-source scenario by controlling various covariates.

### F.3. Real-World Data: Retinal Fundus Images

**Synthetic multi-source Retina**   The following images are also generated when two co-variates interpolately increase from the minimal 30% until the maximal 30%. Figure 13 describes generated retinal fundus images by controlling these three covariates independently. In Figure 13(a) the variable cataract is set to 0 (no cataract) while the age and spherical power are modified between the interval of the minimal 30% and the maximal 30%. The brightness of images changes with the increasing spherical power. In Figure 13(b) generated images with the cases of cataracts are generally more blurry and the vessels can not easily be observed, while the colors of images turn more yellow and grey compared to non-cataract retinal images. Furthermore, Figure 15 presents two more examples of generated retinal images when the age is 59 years old. We can observe in Figure 15(a) and Figure 15(b) that generated images are diverse, even though they belong to the same class.

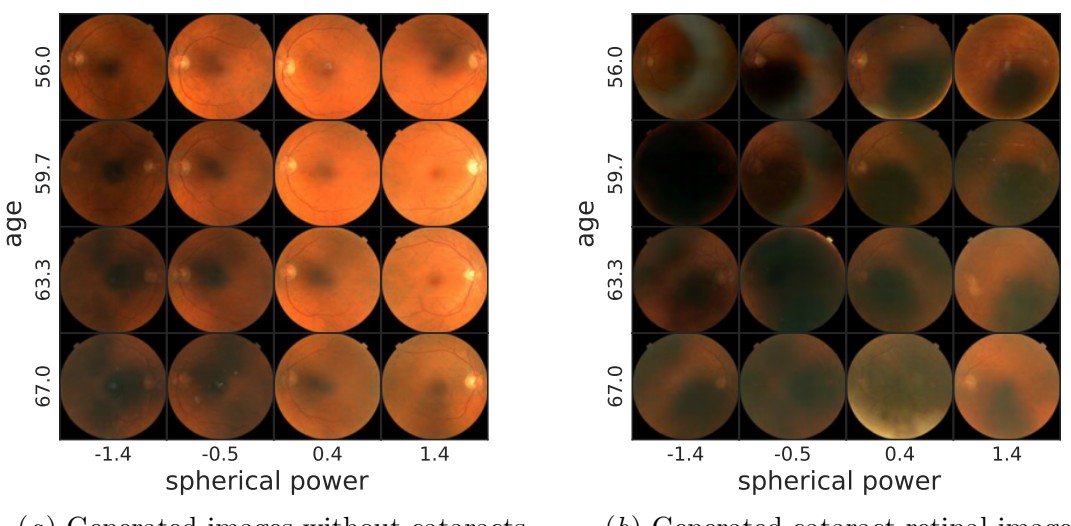

(a) Generated images without cataracts          (b) Generated cataract retinal images

**Figure 13:** Generated retinal fundus images by the "full" multi-source GAN. Here, the model controls ages and spherical powers.

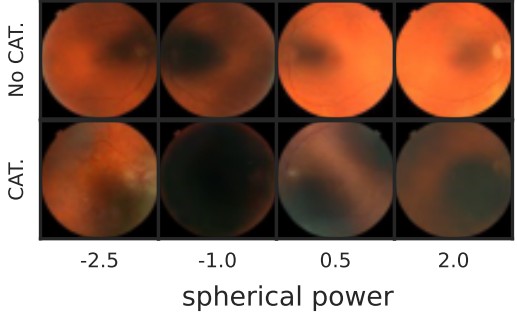

**Figure 14:** Retinal fundus images generated by the "full" multi-source GAN. The model controls cataract and spherical power, age is set to 59. "CAT." represents Cataract.

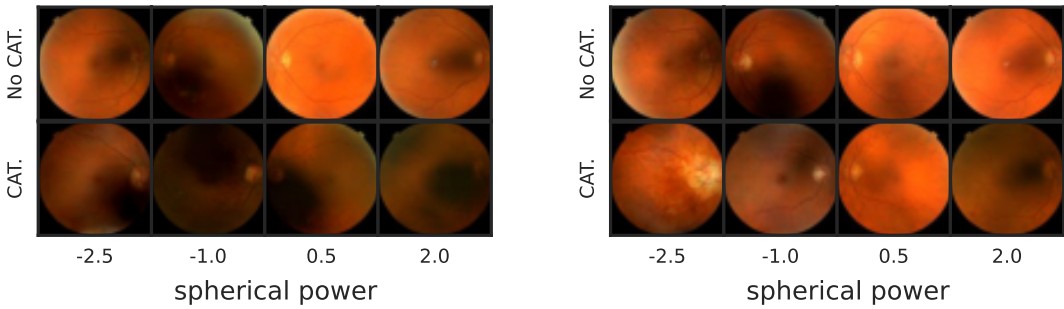

(*a*) Generated retinal images with an age of (*b*) Another example for generated retinal
59 years                           images with an age of 59 years

**Figure 15:** Two different seeds from the "full" multi-source GAN when generating retinal images. Here, the model controls cataracts and spherical powers. Generated images are various, even though they belong to the same class. "CAT." represents Cataract.

