# OpenReview forum: "Heterogeneous Medical Data Integration with Multi-Source StyleGAN"
_MIDL.io/2024/Conference — MIDL 2024 Poster_

### Official Review · Reviewer_WiUs · 2024-02-26

**Confidence:** 4
**Preliminary Rating:** 4
**Recommendation:** Poster
**Final Rating:** 4

**Summary:**

Multi-source Style-GAN (MSSG) is introduced as a conditional generative image model,
showcasing its ability to simultaneously learn from multiple data sources.

The covariates  consist of three parts: a shared part available in both sources (c_shar), a
unique part exclusive to each source (c_uniq), and a hidden part available in one source
but not the other (c_hidd).

Experiments highlight MSSG  can enhance data quality and control variables
across disparate sources, offering potential solutions to data scarcity and label scarcity
issues in medical image data, particularly for rare diseases.
While MSSG relies on a hand-designed latent space model, future work may explore
alternatives like non-parametric models like DECAF and extend the multi-source
integration approach to other generative models like VAEs and diffusion models.

**Strengths:**

The introduction section is written very well. The motivation behind multi-centric data-driven learning is emphasized well

The work is interesting and involves fitting dataset-dependent models to relate covariates which might be very useful to estimate covariates, especially in cases where the person’s habits can play a role. One example is to integrate social factors like habits, and profession to estimate the possibility of a disease, a miscarriage in the case of pregnancy, etc.

**Weaknesses:**

Section 2.2, “The remaining variables are unique to each data source” —---> To improve readability, the authors can quote examples for unique variables.

The approach requires careful handcrafting of parametric models for modeling the relationship between covariates

Sec 2.3 is detailed but adding a training pseudo code in the Appendix will improve the readability far better.

Although the authors provide the reasoning for using MorphoMNIST, having MNIST as the first experiment dilutes the experiment section given such a strong and very interesting introduction for medical images.

**Detailed Comments:**

Can the multi-source technique demonstrated in this study be extended to handle datasets containing three or more sources? Although the authors primarily focused on combining two datasets as multi-source data in their performance evaluation.

Figure 4 illustrates the latent space model of the Multi-Source Style Generative Adversarial Network (MSSG) in the paper. Including this figure in the main part of the paper would greatly benefit readers, providing a clearer understanding of the underlying
concept and architecture.

The authors assumed that the conditional distribution does not shift between datasets. This assumption simplifies the modeling process and is often reasonable in certain contexts; it may not hold for all data. Depending on the nature of the data and the
specific domains being studied, there could be instances where the conditional distribution does shift between datasets due to various factors such as different populations, sampling biases, or changes in experimental conditions. Therefore, it is important for researchers to critically assess the validity of this assumption in their specific application domains and datasets.

Additionally, the authors are encouraged to evaluate the model performance using additional evaluation metrics such as Inception Score and LPIPS Similarity. These metrics provide complementary insights into the quality and diversity of the generated samples, which can help draw more generalized conclusions about the effectiveness of the MSSG model.

**Justification Of Final Rating:**

The authors have addressed most of the comments that were highlighted. The addition of a pseudocode and more evaluation metrics are appreciated. Therefore, I would like to stick to my original rating.

**Justification Of The Preliminary Rating:**

The work is interesting as additional priors based on patient information are taken into account. This boosts the generative capability of the model and is more controllable than complete black-box models

**Questions To Address In The Rebuttal:**

Most existing conditional generative models cannot leverage multiple data sources simultaneously unless label overlap is perfect. —----------> Do we have a citation for this?

In Appendix D, shouldn’t it be E[blood pressure] = ?, instead of blood pressure =?

In sources where data anonymity is of concern? Are there any alternative covariates related to scanner settings, or acquisition settings to consider?

**Special Issue:**

Yes

---

> ### Author Response · Authors · 2024-03-17
> **Response to Reviewer WiUs**
>
> We thank the reviewer for their valuable comments and suggestions on our work and manuscript. We uploaded an updated version of the paper and address their comments point-by-point below.
>
> >Section 2.2, “The remaining variables are unique to each data source” —---> To improve readability, the authors can quote examples for unique variables.
>
> For section 2.2, we added some examples for specific variables, which exist only in some data sets and are not generally collected from every institution. For example, UK Biobank is a more general study, while ADNI is very enriched in individuals with cognitive impairment. Therefore, we used unique variables such as grey matter volumes as unique variables in UKB and hippocampal volumes as the unique variables in ADNI. In the retinal fundus case study, cataract is a disease-specific variable. These variables usually have fewer samples and do not exist in every single data source.
> >The approach requires careful handcrafting of parametric models for modeling the relationship between covariates.
>
> In our experiments, we carefully handcrafted latent space models to approximate missing covariates. This is not an end-to-end solution, but we wanted to make sure that latent variables are fitted well in different sources (More details in appendix D. Parametric Models).
>
> However, the proposed method can be extended with a more general non-parametric model, such as DECAF [1], or also with a straightforward parametric model such as the Gaussian Mixture Model for a more approximate solution.
>
> [1] Boris van Breugel, Trent Kyono, Jeroen Berrevoets, and Mihaela van der Schaar. Decaf: Generating fair synthetic data using causally-aware generative networks. Advances in Neural Information Processing Systems, volume 34, pages 22221–22233. Curran Associates, Inc., 2021.
>
> >Sec 2.3 is detailed but adding a training pseudo code in the Appendix will improve the readability far better.
>
> We thank the reviewer for this suggestion and added a pseudo code of the proposed method to the Section "Multi-Source StyleGAN" in Appendix B to ensure readability.
>
> >Although the authors provide the reasoning for using MorphoMNIST, having MNIST as the first experiment dilutes the experiment section given such a strong and very interesting introduction for medical images.
>
> We used MorphoMNIST as a simulated validation study with known generating distribution to verify that our proposed method works reliably.
> While we agree that the retinal fundus and brain MRI datasets are of much higher application interest, we preferred to prioritize these stronger sanity checks over a slightly more appealing story.
>
> >Can the multi-source technique demonstrated in this study be extended to handle datasets containing three or more sources? Although the authors primarily focused on combining two datasets as multi-source data in their performance evaluation.
>
> Yes, our method can be generalized to $\ge 3$ data sources.
> Depending on which latent variables are available in which data source, this leads to modeling choices within the latent space models.
> E.g., if a variable is available in sources 1 and 2 but not in source 3, we have to decide if we want to sample it from a model derived from source 1 or from source 2, or maybe alternate between the two.
> The mathematical notation would also become more verbose, so we only presented the case of two sources, as this was the focus in our experiments.
>
> Furthermore, we included this part to a small section in the Appendix B to provide a more general aspect of the proposed method.
>
> >Figure 4 illustrates the latent space model of the Multi-Source Style Generative Adversarial Network (MSSG) in the paper. Including this figure in the main part of the paper would greatly benefit readers [...]
>
> We now include the figure of the latent space architecture in the main text and hope that this can help clarify the underlying concept and proposed method when reading Sections 2.2 and 2.3.
> >The authors are encouraged to evaluate the model performance using additional evaluation metrics such as Inception Score and LPIPS Similarity.[...]
>
> We now included more evaluations of our experiments with the suggested LPIPS. However, LPIPS is a pair-wise metric and compares the perceptual similarity of patches between two images. Therefore, we can only use this metric in the controlled multi-source simulation but not in the real-world multi-source experiment, because we do not have the exact labels of both sources. Table 4 in the appendix shows the results of the evaluations from LPIPS, SSIM, and PSNR. Our proposed method outperforms the single-source models in measuring the pairwise similarity between generated and real images, conditioned on the same covariates. Nevertheless, we think that pairwise metrics should be used carefully since images could look different, even though they are conditioned on the same covariates. Please also consider our answer to Reviewer mQtq.

---

> > ### Author Response · Authors · 2024-03-17
> > **Further response to Reviewer WiUs**
> >
> > We further add here two more responses to the reviewer, since there is not enough space left on the official comment above. We are sorry for the inconvenience.
> >
> > >In Appendix D, shouldn’t it be E[blood pressure] = ?, instead of blood pressure =?
> >
> > In this scenario, blood pressure is a continuous variable and sex is a binary variable.
> > In this model, we fit a linear regression model for blood pressure and a logistic regression model for sex. A linear regression model indeed predicts the expected value conditioned on the input variable, but in the stochastic model, the noise variable $\epsilon$ is included: $\texttt{bloodpressure} = f^1(\texttt{age}) = \alpha \texttt{age} + \beta + \epsilon$.
> > We have, however, $\mathbb{E}[\texttt{bloodpressure}] = \alpha \texttt{age} + \beta$.
> > Note the important distinction that $f^j$ is indeed a stochastic function from which we can sample, and not only make a "best-guess" approximation, as described in Section 2.2.
> >
> > >In sources where data anonymity is of concern? Are there any alternative covariates related to scanner settings, or acquisition settings to consider?
> >
> > In the real-world setting of MRI experiments, we have an additional covariate $c^j_\texttt{source}$, $j \in (1, 2)$, which is used to learn data sources. This is trained like a binary variable, which can distinguish between sources (see Appendix C. Implementation Details).
> > Furthermore, if data sources contain different equipment settings, this can be controlled by $c^j_\texttt{source}$ to generate an image that comes from the $j$ cohort.

---

### Official Review · Reviewer_4eEN · 2024-02-27

**Confidence:** 5
**Preliminary Rating:** 3
**Final Rating:** 3.5

**Summary:**

Conditional deep generative models have emerged as powerful tools for generating realistic images, enabling fine-grained control over latent factors. However, in the medical domain, the scarcity of data and the need to integrate information from diverse sources pose significant challenges for existing models. These challenges often result in low-quality image generation and limited controllability. To address these issues, this paper introduces Multi-source StyleGAN (MSSG), a conditional generative image model capable of learning from multiple disparate data sources concurrently. MSSG not only overcomes data scarcity but also addresses heterogeneity by generating new images while controlling all covariates simultaneously. Furthermore, authors conduct extensive experiments on various data, validating MSSG’s ability to control variables from different sources collectively, even without access to paired variables. The proposed algorithm offers a promising direction for unbiased data generation from disparate sources.

**Strengths:**

1. This paper verified the proposed method in different cases: semi-synthetic data of handwritten digit images with varying morphological features,  controlled multi-source simulations on retinal fundus images and brain magnetic resonance images, and a real-world setting of brain MRI from different sources. Through a large number of experiments, it was proved that the system has good performance and robustness.
2. The paper has no obvious grammatical errors.
3. The introduction part of the paper is logical, and the method description is relatively clear.
4. Carefully considered the feasibility of future work and provided research directions for future studies.

**Weaknesses:**

1. Pay attention to essay writing format, the formats of figures and tables in the text should be fixed as uniformly as possible. For example, why is the use of bold text inconsistent in Table 1, Table 2, and Table 3?
2. The authors only list the results of experiments, but the reason is not adequately explained.
3. In this paper, the text's messy layout and lack of visual effects such as graphs or tables make it more difficult for readers to understand and remember the information presented. For example, the proportion of text and images in Figure 3 is not harmonious.
4. The description seems not clear enough, for example, what is the basis for selecting the covariates in the article?

**Detailed Comments:**

1. Pay attention to essay writing format, the formats of figures and tables in the text should be fixed as uniformly as possible. For example, why is the use of bold text inconsistent in Table 1, Table 2, and Table 3?
2. The authors only list the results of experiments, but the reason is not adequately explained.
3. In this paper, the text's messy layout and lack of visual effects such as graphs or tables make it more difficult for readers to understand and remember the information presented. For example, the proportion of text and images in Figure 3 is not harmonious.
4. The description seems not clear enough, for example, what is the basis for selecting the covariates in the article?

**Justification Of Final Rating:**

The authors have already addressed most of my concerns. However, the explanations of the basis for selecting the covariates are still not easy for readers to understand. I hope the authors can provide an in-depth description of this issue in the final revised version.

**Justification Of The Preliminary Rating:**

This paper verified the proposed method in different cases: semi-synthetic data of handwritten digit images with varying morphological features,  controlled multi-source simulations on retinal fundus images and brain magnetic resonance images, and a real-world setting of brain MRI from different sources. Through a large number of experiments, it was proved that the system has good performance and robustness. However, some issues need to be resolved. For example, the authors only list the results of experiments, but the reason is not adequately explained.

**Questions To Address In The Rebuttal:**

1. Pay attention to essay writing format, the formats of figures and tables in the text should be fixed as uniformly as possible. For example, why is the use of bold text inconsistent in Table 1, Table 2, and Table 3?
2. The authors only list the results of experiments, but the reason is not adequately explained.
3. In this paper, the text's messy layout and lack of visual effects such as graphs or tables make it more difficult for readers to understand and remember the information presented. For example, the proportion of text and images in Figure 3 is not harmonious.
4. The description seems not clear enough, for example, what is the basis for selecting the covariates in the article?

---

> ### Author Response · Authors · 2024-03-17
> **Response to Reviewer 4eEN**
>
> We appreciate the reviewer for their valuable feedback. Below, we address your comments and modified our manuscript accordingly.
> If apart from these you have no further concerns, we hope that you will reconsider your rating.
> > Pay attention to essay writing format, the formats of figures and tables in the text should be fixed as uniformly as possible. For example, why is the use of bold text inconsistent in Table 1, Table 2, and Table 3?
>
> We thank the reviewer for pointing this out and we modified the tables and figures in our manuscript to ensure that the formats stay uniform.
>
> >In this paper, the text's messy layout and lack of visual effects such as graphs or tables make it more difficult for readers to understand and remember the information presented. For example, the proportion of text and images in Figure 3 is not harmonious.
>
> The proportion of text and images in Figures is modified. Regarding the layout of our manuscript, we have a lot of experiments, including three different data sets and a real multi-source scenario, and would like to show the results and evaluations in the main manuscript.
>
> The layout of the texts and images in the original Figure 3 is modified and is provided now in the Appendix as Figure 14 in the revised manuscript.
> Furthermore, we put the figure of the model's architecture in the main text to visually clarify our proposed method. With this, we hope that readers can easily understand the concept and architecture.
>
> >The description seems not clear enough, for example, what is the basis for selecting the covariates in the article?
>
> For the MRI use case, we focused on cognitive impairment, a common condition of interest in neuroimaging research.
> In the controlled multi-source setting, age, ventricle volumes, and grey matter volumes have all been prominently shown to be linked to cognitive impairment [1-5].
> In the real-world setting, we additionally selected left/right hippocampal volumes, since they are also related to cognitive impairment.
>
> In the retinal fundus use case, we selected age, cataract, and spherical power as the controlled covariates.
> Cataract is a common condition with a high disease burden. It also leads to a blurring of retinal fundus images, making it a confounding factor, e.g., when classifying other ophthalmological diseases, and hence an interesting target to control.
> We additionally selected spherical power as a proxy for the strength of myopia and hyperopia, which are of universal interest due to their high prevalence.
> Age is also a common risk factor for most eye conditions.
>
> [1] Age-related changes of lateral ventricular width and periventricular white matter in the human brain: A diffusion tensor imaging study. Neural regeneration research
>
> [2 ] Ventricular enlargement as a possible measure of Alzheimer’s disease progression validated using the Alzheimer’s disease neuroimaging initiative database.
>
> [3] Age-related changes in grey and white matter structure throughout adulthood. NeuroImage
>
> [4] Gray Matter Volume and Cognitive Performance During Normal Aging. A Voxel-Based Morphometry Study, Frontiers in Aging Neuroscience
>
> [5] Assessing age-related gray matter decline with voxel-based morphometry depends significantly on segmentation and normalization procedures, Frontiers in Aging Neuroscience

---

### Official Review · Reviewer_mQtq · 2024-03-05

**Confidence:** 3
**Preliminary Rating:** 3
**Recommendation:** Poster
**Final Rating:** 3.5

**Summary:**

This paper proposes a conditional StyleGAN method that combines multiple source datasets with different data attributes to synthesize new images. The synthesized images can be controlled by unified attributes containing attributes from multiple sources. The authors evaluate their method on a simulated multi-source setting on  hand written digit, brain, and fundus imaging datasets, and a real multi-source setting on brain MRIs.

**Strengths:**

- The proposed novel multi-source StyleGAN (MSSG)  method integrates the different attributes from multiple heterogeneous datasets in the unified conditional variable. This enables image generation utilizing all data attributes, addressing the issues of data scarcity and heterogeneity in medical imaging.

- The generated images are evaluated on three simulated settings and one real setting with FID (Frechet Inception Distance) for image quality and a newly proposed metric (strata prediction score) for the controllability of data attributes.

- Synthesized results show a good correlation with ventricle size in Figure 1.

**Weaknesses:**

- The motivation of combining different cohort datasets to generate is suitable to the medical field, but discussion on related works is lacking in synthesis with multiple domains [a-d] and data harmonizations [e-f]

[a] StarGAN v2: Diverse Image Synthesis for Multiple Domains, CVPR’20

[b] Multi-attribute Pizza Generator: Cross-domain Attribute Control with Conditional StyleGAN, BMVC’21

[c] DVG-Face: Dual Variational Generation for Heterogeneous Face Recognition, IEEE PAMI’22

[d] Structure-preserving image translation for multi-source medical image domain adaptation, Pattern Recognition’23

[e] Deep Generative Medical Image Harmonization for Improving Cross-Site Generalization in Deep Learning Predictors, JMRI’21

[f] Style Transfer Using Generative Adversarial Networks for Multi-site MRI Harmonization, MICCAI’21

[g] Harmonizing Flows: Unsupervised MR harmonization based on normalizing flows, IPMI’23



- The proposed method performed better than the single-source StyleGAN baseline. However, it is not compared with any of the multi-source approaches. Some of the above-mentioned baselines [a-f] and a few simple baselines like CycleGANs can be included for fair evaluation.

- I understand the evaluation image generation with non-reference metrics like FID/KID. However, the benefit of medical image synthesis lies in its utilization in downstream tasks rather than solely relying on image quality metrics. Also, is it possible to generate controlled image generation such that it can be evaluated on paired metrics like SSIM or PSNR?

**Detailed Comments:**

Please see weaknesses and also clarify the below questions.

- The proposed new strata prediction score is not common. Could authors explain it clearly so that the evaluation of the controllability of the attribute in Figure 2 is better appreciated?

- The authors claim that the proposed method can address data or label scarcity issues in rare diseases. Please explain how it tackles rare disease cases.

**Justification Of Final Rating:**

I am satisfied with most of the responses from the authors. However, the authors disregard concerns regarding the other multi-source approaches, stating them as image-to-image translation methods. The proposed method is also an image-to-image translation method based on different data attributes, where the characteristics or style of the image is translated conditioned on attributes. Furthermore, the evaluation is weak as it is compared against single source baselines with the FID/KID score (can be unreliable, as pointed out) and strata prediction score (not fully clear). Since the proposed approach is new in combining multi-source attributes, I would adhere to my original score.

**Justification Of The Preliminary Rating:**

This paper proposes a novel way to combine different attributes from multiple heterogeneous sources to generate images. However, it lacks evaluations with multi-source baselines and discussion of relevant multi-source approaches.

**Questions To Address In The Rebuttal:**

The proposed method needs more discussion on relevant literature, evaluations, and clarifications mentioned above.

---

> ### Author Response · Authors · 2024-03-17
> **Response to Reviewer mQtq**
>
> We thank the reviewer for their valuable feedback. We provide a point-by-point response to their concerns and questions below.
> >The motivation of combining different cohort datasets to generate is suitable to the medical field, but discussion on related works is lacking in synthesis with multiple domains and data harmonizations.
>
> Thank you for pointing out the lack of discussion on related work, we added this to the background section A.
> >The proposed method performed better than the single-source StyleGAN baseline. However, it is not compared with any of the multi-source approaches. Some of the above-mentioned baselines and a few simple baselines like CycleGANs can be included for fair evaluation.
>
> We empirically evaluated the proposed method on three different types of datasets and ensured that the proposed multi-source method performed better than the single-source baseline.
> The mentioned works from the reviewer are mostly related to image-to-image translation, where images are transformed from one source to another source (e.g., cats to dogs, horses to zebras, or CT scans to MRI scans).
> Most of these works do not apply to our setting, where we generate images conditional on latent factors that are only partially available in each source.
> >Is it possible to generate controlled image generation such that it can be evaluated on paired metrics like SSIM or PSNR?
>
> For our experiments, we evaluated our proposed method with FID/KID and also the strata prediction score.
> FID and KID evaluate the image quality and do not require paired images. We use the proposed strata prediction score to evaluate the controllability of the models, i.e., whether the generated images faithfully represent the desired input variables.
>
> In our opinion, pairwise metrics are not an optimal evaluation setting for our specific task.
> Images are created from a small set of specified factors, but even for a fixed set of factors a large perceptual variation in the output images is permissible -- the strata prediction score is a better indicator focusing only on the variation due to the latent factors.
>
> Nevertheless, we also added pairwise metrics, namely, SSIM, PSNR, and LPIPS (Table 5 in the appendix).
> In the controlled multi-source settings, we can compare generated images with real test set images, because we have the ground truth labels for each sample.
>
> Our proposed method outperforms the single-source models across all three metrics and all three data sets.
> Note that this evaluation is not possible for the real-world multi-source experiment since we do not have all ground truth covariates for each image.
>
> > Could authors explain it clearly so that the evaluation of the controllability of the attribute in Figure 2 is better appreciated?
>
> The proposed strata prediction score evaluates the controllability of generative models.
> For multiple continuous conditions, we divide the range of each covariate into three quantiles -- for three covariates, this results in $3^3 = 27$ separate strata.
> While each marginal stratum contains exactly a third of all data points, not all joint strata contain exactly $1/27$th of all data points since data needs not be uniformly distributed across the range.
> The strata prediction score then is Pearson's correlation between the predicted values from real images and the generated images within each stratum, averaged over all strata. Therefore, the range of the score is $[-1, 1]$.
> The higher the score is, the better the generated images represent the correct labels.
>
> The idea of the strata prediction score is inspired by the Intra FID (Miyato and Koyama, 2018) [1].
> Intra FID addresses only categorical conditions and the reliability of FID evaluation is highly unstable for smaller sample sizes, which is a big issue when stratifying across multiple factors.
>
> The strata prediction score applies to both categorical and continuous data, is more directly interpretable (as it shows how well each factor is controllable), and is less prone to small sample size (as correlation scores on small samples are more reliable than FID scores).
>
> [1] cGANs with Projection Discriminator, ICLR'18.
>
> > Please explain how it tackles rare disease cases.
>
> Studies with a focus on rare or less common diseases usually have much smaller sample sizes than more general studies, but are also highly enriched for their respective target condition; these dataset sizes tend to be insufficient to reliably train generative image models.
>
> Our approach can increase the total dataset size by joining multiple datasets each with its own enriched set of rare conditions.
> Here, data from other sources effectively function as control cases for the rare trait.
> The larger joint multi-source dataset can enable a better overall generative model, while still representing each rare condition due to its relatively high case count in the enriched source.
> We note, however, that a correct application in real-world settings requires more careful analysis in future work.

---

### Author Response · Authors · 2024-03-17
**General Response**

We would like to thank the reviewers for their thoughtful reviews, which helped us improve the manuscript.
We tried our best to address all comments in the answers to each reviewer individually.

In the main text, we moved the figure with the proposed model's architecture from the appendix to the main text to improve the readability and understanding of Sections 2.2 and 2.3. Furthermore, we added some examples for the unique covariates in section 2.2.
However, we needed to move some results into the appendix to fit the page limit. Furthermore, we improved the formatting of the tables and text carefully.

Regarding the related works, we added a new section "Multi-Source/Multi-Domain Image Generation" to the Section "Background" in Appendix A. Furthermore, we added a pseudo code of the proposed method to the Section "Multi-Source StyleGAN" in Appendix B to ensure readability, and we also added a small subsection "Generalization to more than 2 data sources" to the same section.

Nevertheless, we added new evaluations to our experiments, namely using three new metrics to measure the similarity of generated and real images.
Here, we used LPIPS, SSIM, and PSNR to evaluate three different datasets. These additional results are shown in Appendix E, and our proposed method outperforms the single-source baselines on these three datasets in almost every metric.

We hope that our comments and modifications on the manuscript answer the questions from the reviewers. Detailed responses to individual reviewers are given below.

---

### Author Response · Authors · 2024-03-25
**A kindly reminder**

Dear all reviewers:

We would like to thank all the reviewers for their thoughtful reviews and suggestions, which are invaluable for improving our paper. We would like to kindly remind the reviewers that the close date of the discussion is approaching.
If there are any concerns or follow-up questions, we would welcome further discussion. Additionally, we sincerely hope for continued fruitful discussions with reviewers mQtq and 4eEN, ensuring that our responses address their concerns convincingly.

We have tried our best to address the reviewers' concerns, and we remain available to clarify any lingering doubts or provide additional information.

Thank you once again for your valuable contributions to our manuscript review process.

---

### Author Response · Authors · 2024-03-27

Dear all reviewers:

We would like to kindly remind the reviewers that the close date of the discussion period is today. If there are any concerns or follow-up questions, we will answer them.

---

### Meta-Review · Area_Chair_b7mu · 2024-03-30

**Recommendation:** Accept (Poster)
**Confidence:** 5

**Metareview:**

This submission presents a novel method for generating images by combining attributes from diverse heterogeneous sources. The presented approach demonstrates value, as confirmed by the thorough experimentation on various datasets including semi-synthetic, simulated, and real-world medical images. While one reviewer expresses concerns about the evaluations against multi-source baselines and comparisons to related methods, the authors have provided clarification on the distinction between their technique and image-to-image translation.

Despite doubts over specific evaluation metrics, this work shows promise and explores a unique direction in image generation with controllable attributes. However, the authors could enhance the final version of their paper by providing more context on the selection of covariates and offering deeper insights into the experimental results. One review has been largely disregarded for its lack of depth in the scientific feedback.

For all these reasons, and with respect to the other submissions, the recommendation is towards Acceptance.

---

### Decision · Program_Chairs · 2024-04-05

Accept (Poster)